# BIRC3 RNA Editing Modulates Lipopolysaccharide-Induced Liver Inflammation: Potential Implications for Animal Health

**DOI:** 10.3390/ijms26072941

**Published:** 2025-03-24

**Authors:** Wangchang Li, Duming Cao, Meiyi Shi, Xiaogan Yang

**Affiliations:** Guangxi Key Laboratory of Animal Breeding, Disease Control and Prevention, College of Animal Science & Technology, Guangxi University, Nanning 530004, China; liwangchang@st.gxu.edu.cn (W.L.); 2318301002@st.gxu.edu.cn (D.C.);

**Keywords:** lipopolysaccharide, RNA editing, inflammation, livestock health

## Abstract

Animals and humans are frequently infected by bacteria or exposed to bacterial derivatives in contaminated food, drinking water, or air, which significantly impacts their health. Among these bacterial sources, LPS (lipopolysaccharide) is the primary culprit. While it is widely known that LPS can cause liver inflammation and damage in animals, few studies have investigated this mechanism from the perspective of RNA editing. In this study, we administered LPS to mice via gavage to induce a liver injury model. We then used RNA editing omics approaches (RE-seq) to analyze RNA editing events potentially leading to liver inflammation following LPS administration, aiming to reveal the crucial role of RNA editing in LPS-induced processes. At the RNA editing level, we observed significant differences between the LPS group and the control (CON) group. Specifically, we identified 354 differentially edited genes, with 192 upregulated and 162 downregulated. These differentially edited genes were significantly enriched in pathways related to apoptosis, mTOR signaling, oxidative stress, and Nf-Kappa B signaling. By further integrating gene expression profiles and using a nine-quadrant analysis, we identified an important gene, *Birc3*, which showed significantly higher editing and expression levels in the LPS group. This gene is directly linked to liver inflammation and damage. The RNA editing of *Birc3* represents a significant potential mechanism underlying LPS-induced liver damage, providing a novel approach for addressing animal and human health issues.

## 1. Introduction

Animals and humans are frequently infected by bacteria or exposed to bacterial derivatives in contaminated food, drinking water, or air [1,2]. Endotoxins are a major component of the cell walls of Gram-negative bacteria, with lipopolysaccharide (LPS) being the primary representative [3]. When bacteria die and lyse, LPS is released into the surrounding environment. Due to its strong chemical and thermal stability, LPS can persist under various environmental conditions for extended periods [4]. As the scale and density of livestock farming increase, poor management practices often lead to substandard living conditions for animals, making them more susceptible to bacterial infections and LPS exposure [5,6,7]. Previous studies have shown that LPS infection can lead to severe health issues such as fever, diarrhea, inflammation, hormonal imbalances, and infertility, and in severe cases, it can even lead to death [8,9,10]. Among these bacterial sources, LPS (lipopolysaccharide) is the primary culprit. It is well known that LPS can cause liver inflammation and damage in animals, but few studies have investigated the mechanisms by which LPS affects the liver from the perspective of the RNA transcriptional regulation of gene expression. RNA editing is a crucial post-transcriptional modification process that alters RNA sequences to regulate gene expression. This modification can introduce nucleotide changes, affecting protein function and stability. Investigating the role of RNA editing in LPS-induced liver inflammation not only enhances our understanding of the molecular mechanisms underlying this complex pathological process but also provides new targets for developing therapeutic strategies.

RNA editing is the process of modifying and altering already transcribed RNA molecules to regulate gene expression and function. Unlike DNA, RNA editing involves “fine-tuning” specific RNA sequences to ensure proper protein function. During this process, RNA editing not only changes the sequence of the RNA itself but can also affect protein synthesis, thereby further regulating various biological functions of the cell. RNA editing is not a random process but is regulated by specific enzymes and mechanisms, which in turn influences gene expression and protein function [11,12]. This process is observed broadly across various organisms, including plants [13], animals [14], and microorganisms [15], where it contributes significantly to processes like adapting to environmental changes [16], responding to stress [17], and regulating metabolism [18]. Common types of RNA editing include adenosine-to-inosine (A-to-I) editing [19,20] and cytidine-to-uridine (C-to-U) editing [21]. A-to-I editing is primarily mediated by a family of enzymes called *ADARs* (adenosine deaminases acting on RNA). *ADAR* enzymes recognize adenine (A) residues in RNA and convert them to inosine (I). This process is crucial for normal cellular function because inosine behaves differently from adenine within RNA, playing a key role in RNA secondary structure formation and subsequent translation into proteins [22]. This type of editing not only affects RNA stability but can also alter how RNA interacts with other molecules, thereby regulating gene expression and function. In addition to A-to-I editing, C-to-U editing is another form of RNA editing, although it has been less studied in vivo. C-to-U editing involves the conversion of cytosine to uracil, affecting RNA structure and function, and thereby regulating protein synthesis in certain contexts [23]. RNA editing currently exhibits broad application prospects. However, accurately quantifying RNA editing events has been challenging due to the lack of standardized methodologies, making cross-study comparisons difficult. To address these limitations, we developed RE-seq [24], a comprehensive computational framework designed for the systematic quantification and differential analysis of RNA editing events in bulk RNA-seq data. Given the maturity of LPS-induced liver injury models, it is hoped that RNA editing sequencing (RE-seq) techniques can be used to identify key targets responsible for LPS-induced liver damage.

Liver injury is often closely linked to cellular apoptosis. *BIRC3* is an anti-apoptotic protein and belongs to the family of inhibitor of apoptosis proteins (IAPs), which includes eight human IAPs that help cells evade apoptosis [25]. Mechanistically, both *BIRC3* and *BIRC2* function as E3 ubiquitin ligases. Additionally, upon receptor-induced activation, *BIRC2/3* forms complexes with TNF receptors 2 or 3 (TNFR2/3), subsequently mediating the ubiquitin-dependent degradation of Nf-Kappa B inhibitors, ultimately leading to the activation of the Nf-Kappa B signaling pathway [26]. *BIRC3* plays a crucial role in regulating the Nf-Kappa B and TNF signaling pathways, which are primary regulators of inflammatory responses [27]. Studies have shown that the knockdown of *BIRC3* gene expression significantly reduces the activity of the Nf-Kappa B pathway in hepatocellular carcinoma (HCC) [28]. Other studies have demonstrated that the high expression of the *BIRC3* gene is often associated with elevated levels of pro-inflammatory cytokines *IL-1b*, *TNF-a*, and *IL-6* in mice [29]. In cancer prognosis research, loss-of-function mutations or deletions in the *BIRC3* gene are consistently associated with shorter survival and poor prognosis in cancer patients [30].

In this study, we aim to elucidate the regulatory mechanisms of RNA editing in LPS-induced liver damage in animals, with a particular focus on its impact on liver inflammation. To achieve this, we conducted comprehensive transcriptomic analyses on feed samples from mice treated with 50 μmol/L LPS (*n* = 3) and untreated control groups (*n* = 3). High-throughput transcriptome sequencing data from both LPS-treated and control samples were preprocessed to identify RNA editing sites and gene expression levels. Subsequently, differential RNA editing (DRE) analysis, differential expression gene (DEG) analysis, Gene Ontology (GO) enrichment analysis, Kyoto Encyclopedia of Genes and Genomes (KEGG) enrichment analysis, and correlation analysis between RNA editing sites and gene expression levels were performed to identify candidate genes associated with LPS-mediated RNA editing leading to inflammation. A summary of this pipeline is shown in the Figure 1. This research provides new insights into the role of RNA editing in the inflammatory pathways triggered by LPS in livestock animal health. It highlights that RNA editing could be a significant area for managing inflammation-related health issues in animals, paving the way for future studies on RNA editing modulators and their biotechnological applications.

## 2. Results

### 2.1. Comprehensive Map of RNA Editing Events

The characteristics of identified RNA editing sites were summarized. RNA editing types were categorized across all samples. C-to-U (C→T or G→A) and A-to-I (A→G or T→A) editing events were predominant, collectively accounting for approximately 80% of all detected editing events. Other editing types, such as A-to-C and A-to-T, were less frequent, reflecting the diversity of the RNA editing landscape (Figure 1A). The majority of RNA editing events were located in exonic regions, underscoring their potential impact on protein-coding genes. Other notable proportions were found in upstream, downstream, splicing, and ncRNA intronic regions, suggesting that RNA editing may influence gene regulation and function at multiple genomic levels (Figure 1B). The analysis of RNA editing-induced SNVs revealed that synonymous SNVs were the most prevalent, followed by nonsynonymous SNVs, which can directly affect protein structure and function. Stop-gain and stop-loss variants, although rare, have profound effects by introducing or removing stop codons, potentially altering protein length and functionality (Figure 1C).

### 2.2. Differential Analysis of RNA Editing Events

The principal component analysis (PCA) of RNA editing frequencies shows distinct clustering of LPS samples and CON samples, highlighting a clear separation along the PC1 (22.56%) and PC2 (20.12%) axes. This indicates that LPS induces significant global changes in RNA editing patterns (Figure 2A). The editing frequency distribution was analyzed across all samples. The histograms revealed a broad range of editing frequencies in both the CON and LPS groups, indicating baseline RNA editing activity under normal conditions in the CON group and altered activity in response to LPS treatment (Figure 2B). The overlaps and unique components of the RNA editing events were assessed using a Venn diagram (Figure 2C). A total of 2287 RNA editing events were common across the CON and LPS groups, indicating that at the RNA editing event level, there are distinct regulatory mechanisms operating between the two groups.

To identify differences in RNA editing genes between LPS and CON samples, we performed differential analysis at both the RNA editing site level and the gene level. A total of 354 RNA editing genes were found to exhibit significant differences (adjusted *p*-value < 0.05) between the LPS and CON groups (Appendix A). Specifically, these differences comprised 192 upregulated RNA editing genes and 162 downregulated RNA editing genes (Figure 3A). This analysis highlights a complex regulatory landscape where RNA editing events within genes can exhibit divergent responses to LPS administration. For instance, certain genes may show increased editing activity, potentially leading to altered protein function or expression levels, while others may display reduced editing activity, which could also impact their biological roles (Figure 3B). These findings suggest that RNA editing plays a crucial role in the cellular response to LPS-induced inflammation.

### 2.3. Functional Enrichment Analysis of Differential Editing Gene Sets

To elucidate the biological implications of RNA editing changes under LPS administration, we performed Gene Ontology (GO) and Kyoto Encyclopedia of Genes and Genomes (KEGG) enrichment analyses on the differentially edited genes at both the gene and editing site levels. These analyses revealed distinct functional categories and pathways related to immune responses and apoptosis associated with each group (Figure 4).

Through Gene Ontology (GO) biological process (BP) pathway analysis, we discovered that the main enriched biological functions were oxidative stress, cell apoptosis, and amino acid metabolism (Figure 4A and Appendix A). BP pathways included the regulation of apoptotic signaling pathway (GO:2001233), nucleotide metabolic process (GO:0009117), and glutathione metabolic process (GO:0006749). A comparative view of KEGG pathway enrichments for RNA editing genes is shown in Figure 4B (Appendix A). RNA editing genes at the gene level showed enrichments in pathways related to glutathione metabolism (mmu00480), PPAR signaling (mmu03320), citrate cycle (TCA cycle) (mmu00020), apoptosis (mmu04210), and mTOR signaling (mmu04150). The GSEA enrichment analysis indicated that the Nod-like receptor signaling pathway was significantly enriched in the LPS-treated group. This pathway is closely associated with inflammation (Figure 4C). The distinct patterns observed in the Gene Ontology (GO) enrichment analysis, Gene Set Enrichment Analysis (GSEA), and Kyoto Encyclopedia of Genes and Genomes (KEGG) enrichment analysis provide a comprehensive view of the potential impacts of RNA editing on apoptosis, oxidative stress, inflammation, and metabolic pathways.

### 2.4. LPS Alters Gene Expression at the Transcriptional Level

High-throughput transcriptome sequencing was performed using the livers of CON and LPS samples. We first calculated the correlation of gene expression among individuals (Figure 5A), and the results showed higher correlation of gene expression between individuals from the same group. Next, a differential analysis was performed to identify DEGs (differentially expressed genes) between LPS and CON. We identified a total of 112 genes, of which 31 were upregulated and 81 were downregulated in LPS compared to CON (Figure 5B).

Through Gene Ontology (GO) Biological Process (BP) pathway analysis for DEGs, we discovered that the main enriched biological functions were T cell activation and cell apoptosis (Figure 5C and Appendix A). BP pathways included alpha-beta T cell activation (GO:0046631) and cell killing (GO:0001906). A comparative view of KEGG pathway enrichments for DEGs is shown in Figure 5D (Appendix A). The KEGG pathway showed enrichments in pathways related to Hepatitis C (mmu05160), IL-17 signaling (mmu04657), Toll-like receptor signaling (mmu04620), TNF signaling (mmu04668), Glycosaminoglycan degradation (mmu00531), and cAMP signaling (mmu04024). The GSEA enrichment analysis indicated that the Nf−Kappa B signaling pathway was significantly enriched in the LPS-treated group. This pathway is closely associated with inflammation (Figure 5E). The distinct patterns observed in the Gene Ontology (GO) enrichment analysis, Gene Set Enrichment Analysis (GSEA), and Kyoto Encyclopedia of Genes and Genomes (KEGG) enrichment analyses provide a comprehensive view of the potential impacts of RNA expression on apoptosis, oxidative stress, inflammation, and metabolic pathways.

### 2.5. Integrated Analysis of RNA Editing and RNA Expression

To comprehensively analyze the relationship between RNA editing and RNA expression, we generated a nine-quadrant scatterplot integrating both data types (Figure 6A). The *x*-axis represents changes in RNA expression (log2 fold change), while the *y*-axis denotes changes in RNA editing levels (log2 fold change in mean editing levels). Genes were categorized into nine regions based on their expression and editing patterns, highlighting those with consistent upregulation or downregulation in both metrics.

Notably, genes involved in inflammation pathways were prominently located in the top-right quadrant, indicating coordinated upregulation in both RNA expression and RNA editing levels. The detailed analysis of RNA editing loci and RNA expression revealed significant differences between CON and LPS samples, particularly in inflammation-related genes such as *Birc3* (Baculoviral IAP Repeat Containing 3) (Figure 6B and Figure 6C). These findings suggest that LPS administration induces coordinated changes in both RNA expression and editing, which may play crucial roles in modulating inflammatory responses. To further investigate the regulatory mechanisms, a linear regression analysis was conducted between the RNA editing levels of key genes and the RNA expression of relevant RNA editing enzymes (Figure 6D). Notably, a strong positive correlation was observed between the RNA expression of Apobec1 (Apobec1 RNA editing enzyme) and the mean allele frequency (MAF) of editing sites in *Birc3* (r = 0.99; *p* < 0.0001). This finding provides evidence that the Apobec1 RNA editing enzyme, upregulated under LPS administration, may mediate the precise editing of inflammatory-related genes, potentially contributing to liver damage. The increased activity of Apobec1 could result in altered RNA editing patterns that affect the expression and function of genes involved in inflammation, thereby exacerbating liver injury.

## 3. Materials and Methods

### 3.1. Ethics Statement

All experimental work was conducted in accordance with national and international guidelines. The protocol for this study was approved by the Animal Care and Welfare Committee of the College of Animal Science and Technology at Guangxi University (Approval Code: GXU2019-021).

### 3.2. Sample Collection and Sequencing

In this experiment, oral administration was used. The control group received saline (CON), while the LPS group received 50 μmol/L LPS diluted in saline (LPS). Each group consisted of three 3-month-old mice. The experimental period lasted one week, with daily dosing. During this period, all mice were maintained under consistent feeding conditions and kept in the same living environment.

Total RNA of the liver in each group was extracted using the TRIzol^®^ RNA extraction reagent (Invitrogen, Carlsbad, CA, USA) according to the manufacturer’s instructions. The integrity of RNA was examined by agarose gel electrophoresis, and the concentration of RNA was measured using NanoDrop 2000 spectrophotometers (Thermo Fisher Scientific, Boston, MA, USA). mRNA was purified by oligo (dT) magnetic beads and fragmented into short fragments using fragmentation buffer. First-strand cDNA was synthesized with SuperScript II Reverse Transcriptase (Invitrogen) using random primers, and second-strand cDNA was synthesized using the synthesized first strand of cDNA as a template. The obtained double-stranded cDNA was purified by a VAHTS^®^ mRNA-seq V3 Library Prep Kit for Illumina (Vazyme, Nanjing, China), end-repaired, poly(A)-added, and then ligated to Illumina sequencing adapters. Sequencing was performed on the Illumina HiSeq-2000 platform for 50 cycles. High-quality reads passed through the Illumina quality filter were retained in fastq.gz format for sequence analysis.

### 3.3. RNA-Seq Data Preprocessing

To ensure high-quality data, raw sequencing reads were processed using FastQC for quality assessment, followed by the removal of low-quality reads.

### 3.4. Alignments and RNA Editing Event Identification

The clean reads were aligned to the reference genome (mm10) using HISAT2 (v0.7.17) with default settings [31]. Then, Samtools (v1.9), Picard tools (v3.1.1), and GATK (v4.0) were used for SNP detection [32,33]. All detected SNPs underwent filtering through the “Variant Filtration” module of GATK, using the following standard parameters: variants with Quality Depth (QD) < 2; FS (Phred-scaled *p*-value using Fisher’s exact test for strand bias detection) > 60; MQRankSum (Z-score of the rank sum of the Phred-scaled mapping qualities) < −12.5; ReadPosRankSum (Z-score of the rank sum of the Phred-scaled position bias estimations) < −8; MQ (root mean square of the mapping quality) < 40.0; the mean sequencing depth of variants (across all individuals) was limited to less than 1/3× and more than 3×; and SOR (strand odds ratio) > 3.0. The following filtering criteria were used: base quality ≥25; sequencing depth ≥5; alternative allele depth ≥3; and editing frequency between 10% and 100%. High-confidence editing events were retained if their editing levels were ≥10% and SNPs were limited to two alleles.

The SNPs obtained through this series of filtering processes are considered to have reliable RNA editing site information. The proportion of reads supporting an RNA editing site relative to the total reads at that site is used as the frequency of the RNA editing event.

### 3.5. Differential Editing Analysis of Differential Editing Events

For each RNA editing site’s reads, we calculated the number of editing event reads in both the LPS and CON groups. Then, based on the RNA editing sites mapped to genes in the mouse reference genome, the gene to which an RNA editing site is mapped is defined as the RNA editing gene for that site. At the gene level, we aggregated the total editing read counts across all sites within a gene to quantify the extent of RNA editing for that gene. We used DESeq2 [34] to analyze differential RNA editing at the gene level, with an adjusted *p*-value < 0.05 considered significant for differential RNA editing genes. Through this differential analysis, we were able to identify which genes showed differences in RNA editing levels between the LPS and CON groups, allowing us to focus on these differentially edited genes.

### 3.6. RNA Expression Quantification and Analysis

We used StringTie (version 2.2.0) [35] to quantify gene expression levels, reconstruct transcripts, and calculate TPM (Transcripts Per Million) values. To identify differentially expressed genes (DEGs), we employed the DESeq2 tool with a threshold of FDR (false discovery rate) < 0.05 and |log2FC| ≥ 1 (absolute log2 fold change greater than or equal to 1). Using this approach, we were able to identify which genes showed significant differences in expression levels between the LPS and CON groups. These differentially expressed genes can help us focus on those that play important roles in liver physiology, thereby further investigating their impact on the liver.

### 3.7. Correlation Analysis Between RNA Editing and Gene Expression

We conducted a multi-omics integrated analysis to explore the relationship between RNA editing genes and gene expression. We compared differentially edited genes with differentially expressed genes. This analysis aims to jointly guide the connection between RNA gene editing and RNA gene expression, providing a screening method for target genes in liver injury. Specifically, we performed quadrant analysis and classification on differentially edited and differentially expressed genes, selecting genes from the quadrants with the highest reliability as candidate genes. Pearson correlation coefficients were used for correlation analysis. The entire workflow was executed in R (version 4.2.0).

### 3.8. Pathway Enrichment

Genes often work in concert to perform specific biological functions. Pathway-based analysis is a valuable approach for understanding the roles of genes in these complex processes. The KEGG (Kyoto Encyclopedia of Genes and Genomes) database [36] stands as one of the foremost publicly accessible resources for pathway-related data. To identify significantly enriched metabolic and signal transduction pathways among CAGs (Candidate-Associated Genes) relative to the entire genome context, pathway enrichment analysis was performed. The method for calculating enrichment is consistent with that employed in Gene Ontology (GO) [37] analysis:P=1−∑i=0m−1 (Mi)(N−Mn−i)(Nn)

In this context, *N* signifies the total count of genes with KEGG annotations, while n denotes the number of CAGs within *N*. *M* represents the total number of genes annotated to particular pathways, and m is the number of CAGs in *M*. Following the calculation of the *p*-value, it was corrected using false discovery rate (FDR) adjustment, with an FDR value of 0.05 or less being set as the threshold. Pathways that meet this criterion are categorized as significantly enriched pathways in CAGs. Finally, we utilize the String database to identify genes that are significantly represented in pathways and create a protein–protein interaction map. Gene Ontology (GO) and Kyoto Encyclopedia of Genes and Genomes (KEGG) analysis were performed using the OmicShare tools, a free online platform for data analysis (http://www.omicshare.com/tools, accessed on 1 January 2024).

Gene Set Enrichment Analysis (GSEA) [38] was performed to identify significant pathways and biological processes affected by LPS administration. Briefly, gene expression data were pre-ranked based on their differential expression significance between the LPS-treated and control groups. The GSEA software (version 4.3.3) from the Broad Institute was used with default settings. Enriched gene sets were considered statistically significant at a false discovery rate (FDR) < 0.05.

### 3.9. Statistical Analysis

Statistical analyses were performed using the SPSS 18.0 software package (SPSS Science, Chicago, IL, USA). Experimental data were subjected to *t*-test and ANOVA analyses, with a significance threshold set at *p* < 0.05. Graphs were generated using GraphPad Prism 8 software (GraphPad, Santiago, MN, USA). Data were presented as mean ± standard deviation (SD).

## 4. Discussion

### 4.1. LPS-Induced Alterations in RNA Editing and Gene Expression: Implications for Liver Inflammation and Damage

LPS, or lipopolysaccharide, is one of the primary components of the outer membrane of Gram-negative bacteria. Scientific research has demonstrated that LPS can have multiple negative effects on animal health [39,40]. It primarily influences the health status of organisms by initiating inflammatory responses. When LPS enters an animal’s body, it activates the immune system, resulting in a range of physiological reactions, including fever, increased white blood cell count, and the production of acute phase proteins. These responses are part of the body’s defense mechanism but can also lead to systemic inflammation and other adverse health outcomes if the response is excessive or prolonged [41,42,43].

We observed that under LPS conditions, the liver RNA editing events in the LPS group exhibited significant differences in RNA regulation compared to the CON group. These differences were evident in the number and frequency of RNA editing events, as well as through PCA (principal component analysis). Further analysis of significantly enriched RNA editing genes revealed several interesting signaling pathways, including glutathione metabolism (mmu00480), PPAR signaling (mmu03320), citrate cycle (TCA cycle) (mmu00020), apoptosis (mmu04210), and mTOR signaling (mmu04150). These pathways are either directly or indirectly associated with liver damage, indicating that RNA editing might play a crucial role in LPS-induced liver injury. This finding suggests a novel angle for understanding the mechanisms underlying LPS-induced hepatic dysfunction.

We also sought to examine changes in gene expression levels under LPS induction. Interestingly, liver-related gene expression in the LPS group exhibited significant differences compared to the CON group. These differences were clearly demonstrated through sample gene expression correlations and differential gene volcano plots. Further analysis of significantly enriched RNA expression genes revealed several key immune, inflammation, and apoptosis signaling pathways, including Hepatitis C (mmu05160), IL-17 signaling (mmu04657), Toll-like receptor signaling (mmu04620), TNF signaling (mmu04668), Glycosaminoglycan degradation (mmu00531), and cAMP signaling (mmu04024). Additionally, GSEA indicated the significant enrichment of the Nf-Kappa B signaling pathway. These pathways clearly illustrate that under LPS administration, the expression patterns of inflammation-related genes are significantly altered, leading to liver inflammation and associated physiological effects such as tissue damage. This suggests that LPS-induced changes in gene expression play a crucial role in the development of liver inflammation and subsequent injury.

### 4.2. Birc3 Editing May Be a Potential Inflammatory Mechanism Induced by LPS

To investigate the regulatory mechanisms of RNA editing and RNA expression, we utilized a nine-quadrant analysis method. Our goal was to uncover the patterns of gene regulation related to RNA editing and RNA expression. Through this analysis, we discovered a regulatory gene, *Birc3*, which is involved in inflammation. Notably, both the editing and expression levels of *Birc3* were significantly higher in the LPS group compared to the control. This suggests that the *Birc3* gene may be a key factor in LPS-induced liver inflammation and damage.

*Birc3* is a multifunctional protein that primarily functions by inhibiting apoptosis, regulating inflammatory responses, and modulating immune reactions. It plays a critical role in various physiological and pathological processes, particularly in the mechanisms underlying cancer and inflammatory diseases [44,45,46]. *Birc3* plays a significant role in inflammatory responses by regulating the NF-kB (Nuclear Factor Kappa-light-chain-enhancer of activated B cells) signaling pathway, promoting the expression of pro-inflammatory cytokines such as TNF-a and IL-6 [47,48,49]. The overactivation of *Birc3* may be associated with chronic inflammatory diseases, such as rheumatoid arthritis and inflammatory bowel disease, as it can continuously activate the NF-kB signaling pathway, leading to sustained inflammation [50].

### 4.3. Strategies for Preventing and Managing LPS-Induced Liver Damage

To address potential liver damage caused by LPS (lipopolysaccharide), we are considering the following partial solutions. First, the use of probiotics can help maintain gut microbial balance, reducing the proliferation of harmful bacteria and thereby decreasing LPS release [51]. Second, increasing dietary fiber intake can promote gut health and reduce LPS absorption [52]. Additionally, developing and applying specific inhibitors of the NF-κB signaling pathway can directly block key components of the inflammatory response [53,54]. Finally, exploring the regulation of RNA editing to restore normal gene expression patterns may help alleviate inflammatory responses [55,56].

### 4.4. Limitations and Future Research Directions

A limitation of this study is that if we could induce BIRC3 RNA editing in the L-O2 cell line or construct an LPS-treated L-O2 cell line and then use RNA editing enzyme interference to observe the impact of BIRC3 RNA editing on liver cell lines, it would significantly enhance our understanding of the role of the BIRC3 gene in liver injury mechanisms. On the other hand, our basic research has revealed the potential role of the BIRC3 gene editing in liver injury but has not provided an effective solution. Future research should further validate BIRC3 RNA editing phenomena in the L-O2 cell line and use RNA editing enzyme interference techniques to explore its specific impacts on liver cell functions. It should also utilize animal models (such as mice) for in vivo experiments to validate in vitro findings and explore the actual effects and mechanisms of BIRC3 RNA editing in living organisms. Finally, it should investigate how RNA editing technologies can provide protection against liver injury and develop efficient therapeutic strategies, for example, using CRISPR-Cas13 or other RNA editing tools to regulate the expression levels or editing status of BIRC3, thereby alleviating liver injury.

## 5. Conclusions

In this study, we constructed an LPS-induced liver injury model and used RNA editing omics approaches (RE-seq) to analyze RNA editing events potentially leading to liver inflammation following LPS administration. Our results reveal the crucial role of RNA editing in LPS-induced liver inflammation and damage, providing new insights into this complex pathological process. Firstly, we observed significant differences in RNA editing between the LPS group and the control (CON) group. Specifically, we identified 354 differentially edited genes, with 192 upregulated and 162 downregulated. These differentially edited genes were significantly enriched in pathways related to apoptosis, mTOR signaling, oxidative stress, and NF-kB signaling. The dysregulation of these pathways has been widely recognized as a key factor contributing to liver inflammation and damage. By further integrating gene expression profiles and using a nine-quadrant analysis, we identified an important gene, *Birc3*. In the LPS group, *Birc3* exhibited significantly higher editing and expression levels. As an anti-apoptotic protein, *Birc3*’s overactivation is closely associated with various inflammatory responses and immune regulation. Our study shows that the high editing and high expression of *Birc3* are directly linked to liver inflammation and damage, indicating that it may be a critical mechanism underlying LPS-induced liver injury.

## Data Availability

Contact the author for data requests if necessary.

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
