# Peer review of "BIRC3 RNA Editing Modulates Lipopolysaccharide-Induced Liver Inflammation: Potential Implications for Animal Health"

_ijms, 2025, doi:10.3390/ijms26072941_

Round 1
Reviewer 1 Report
Comments and Suggestions for Authors
This study has an interesting idea. I have following concerns.
Please define RNA editing.
Please define RNA editing genes (Line 209).
Please define RNA editing frequencies (Line 313), and provide detailed RNA sequence changes between LPS and control.
Please define RNA editing omics approaches (RE-seq) at Lines 15 and 48, and provide a reference to support.
Please provide a workflow diagram to illustrate this new technology in detail.
Materials: provide mouse information of animal, age, male or female for this study?
Sequential order # at lines 102 and 115 is overlapping.
Line 199: (Figure 3C). Please check: not Figure 2C?
Line 199: A total of 2,257 RNA editing. Please check: Figure 2C shows 2287?
Please describe 354 RNA editing genes (Line 211) in details. Are these genes specialized in editing RNA, or are they genes with RNA edits, or are they differentiated genes between LPS and controls? If these 354 genes are important for editing RNA sequence, please list all these genes in a supplementary table.
Comments on the Quality of English Language
Please have English and figure data proofread.
Author Response
Thank you very much for your valuable feedback. Your suggestions have greatly improved the research prospects and overall quality of our manuscript. We have carefully addressed each of your comments and have made the necessary revisions. Our detailed responses to your feedback are provided in the attached document. We appreciate your attention to this matter.

Reviewer 2 Report
Comments and Suggestions for Authors
Comments
The manuscript with entitled “ BIRC3 RNA Editing Modulates LPS-Induced Liver Inflamma-2 tion: Potential Implications for Animal Health”.
- This manuscript discusses the role of RNA editing in LPS-induced liver inflammation and potential implications for animal health. The article found a significant change of BIRC3 RNA editing.
- The manuscript is original. The article provides materials for research in this field.
- In the introduction, the author should introduce The author should introduce the meaning of RNA editing and its significance in LPS-induced liver inflammation.
- The author applied multiple analytical methods. How effective are these methods in this article? What are the successful indicators used in these methods? The author should introduce.
- In the conclusions, the author found that BIRC3 RNA editing has significant changes in LPS-induced, and the author should focus on introducing the role of BIRC3.
What are the specific limitations of the article and future research directions? The author should focus on discussing?
- The format of the references is confusing, and it does not meet the requirements of the journal.
Author Response

(The authors gave the same response as above.)

Round 2
Reviewer 1 Report
Comments and Suggestions for Authors
No further comments. Congratulations.
Reviewer 2 Report
Comments and Suggestions for Authors
None